

# Characterization of the salivary microbiome in patients with pancreatic cancer

Pedro J. Torres[1], Erin M. Fletcher[2], Sean M. Gibbons[3,4], Michael Bouvet[5], Kelly S. Doran[1,6] and Scott T. Kelley[1]

[1] Department of Biology, San Diego State University, San Diego, CA, United States
[2] Department of Medical Sciences, Harvard University, Boston, MA, United States
[3] Graduate Program in Biophysical Sciences, University of Chicago, Chicago, IL, United States
[4] Institute for Genomics and Systems Biology, Argonne National Laboratory, Lemont, IL, United States
[5] Department of Surgery, University of California, San Diego, La Jolla, CA, United States
[6] Department of Pediatrics, University of California San Diego School of Medicine, La Jolla, CA, United States

Corresponding author
Scott T. Kelley,
skelley@mail.sdsu.edu

## ABSTRACT

Clinical manifestations of pancreatic cancer often do not occur until the cancer has undergone metastasis, resulting in a very low survival rate. In this study, we investigated whether salivary bacterial profiles might provide useful biomarkers for early detection of pancreatic cancer. Using high-throughput sequencing of bacterial small subunit ribosomal RNA (16S rRNA) gene, we characterized the salivary microbiota of patients with pancreatic cancer and compared them to healthy patients and patients with other diseases, including pancreatic disease, non-pancreatic digestive disease/cancer and non-digestive disease/cancer. A total of 146 patients were enrolled at the UCSD Moores Cancer Center where saliva and demographic data were collected from each patient. Of these, we analyzed the salivary microbiome of 108 patients: 8 had been diagnosed with pancreatic cancer, 78 with other diseases and 22 were classified as non-diseased (healthy) controls. Bacterial 16S rRNA sequences were amplified directly from salivary DNA extractions and subjected to high-throughput sequencing (HTS). Several bacterial genera differed in abundance in patients with pancreatic cancer. We found a significantly higher ratio of *Leptotrichia* to *Porphyromonas* in the saliva of patients with pancreatic cancer than in the saliva of healthy patients or those with other disease (Kruskal–Wallis Test; $P < 0.001$). *Leptotrichia* abundances were confirmed using real-time qPCR with *Leptotrichia* specific primers. Similar to previous studies, we found lower relative abundances of *Neisseria* and *Aggregatibacter* in the saliva of pancreatic cancer patients, though these results were not significant at the $P < 0.05$ level (K–W Test; $P = 0.07$ and $P = 0.09$ respectively). However, the relative abundances of other previously identified bacterial biomarkers, e.g., *Streptococcus mitis* and *Granulicatella adiacens*, were not significantly different in the saliva of pancreatic cancer patients. Overall, this study supports the hypothesis that bacteria abundance profiles in saliva are useful biomarkers for pancreatic cancer though much larger patient studies are needed to verify their predictive utility.

## INTRODUCTION

In the United States, approximately 40,000 people die every year from pancreatic adenocarcinoma, making it the fourth leading cause of cancer related death. Patients diagnosed in the early stage of pancreatic cancer have a 5-year survival rate of 24%, compared to 1.8% when diagnosed in the advanced stage (*Li et al., 2004*). Clinical manifestations of pancreatic cancer do not appear until after the cancer has undergone metastasis (*Holly et al., 2004*), emphasizing the need for early detection biomarkers. The etiology of pancreatic cancer remains elusive, with cigarette smoking being the most established risk factor (*Vrieling et al., 2010*; *Nakamura et al., 2011*; *Fuchs, Colditz & Stampfer, 1996*; *Zheng et al., 1993*), although links have also been made to diabetes (*Haugvik et al., 2015*; *Liu et al., 2015*), obesity (*Bracci, 2012*), and chronic pancreatitis (*Malka et al., 2002*). Recent research has also shown that men with periodontal disease have a two-fold greater risk of developing pancreatic cancer after adjusting for smoking, diabetes, and body mass index (*Michaud et al., 2007*).

The human oral cavity harbors a complex microbial community (microbiome) known to contain over 700 species of bacteria, more than half of which have not been cultivated (*Aas et al., 2005*). Researchers have identified a core microbial community in healthy individuals (*Zaura, Keijser & Huse, 2009*) and shifts from this core microbiome have been associated with dental carries and periodontitis (*Berezow & Darveau, 2011*). The composition of bacterial communities in saliva seems to reflect health status under certain circumstances (*Yamanaka et al., 2012*), making the analysis of salivary microbiomes a promising approach for disease diagnostics. A study by *Mittal et al. (2011)* found that increases in the numbers of *Streptococcus mutans* and lactobacilli in saliva have been associated with oral disease prevalence, while another study showed that high salivary counts of *Capnocytophaga gingivalis*, *Prevotella melaninogenica* and *Streptococcus mitis* may be indicative of oral cancer (*Mager et al., 2005*).

A recent study by *Farrell et al. (2012)* suggested that the abundances of specific salivary bacteria could be used as biomarkers for early-stage pancreatic cancer. Using the Human Oral Microbe Identification Microarray (HOMIM), researchers observed decreased levels of *Neisseria elongata* and *Streptococcus mitis* in patients with pancreatic cancer compared with healthy individuals, while levels of *Granulicatella adiacens* were significantly higher in individuals with pancreatic cancer (*Farrell et al., 2012*). The HOMIM's ability to detect 300 of the most prevalent oral bacterial species has made it a suitable method for assessing community profiles at the phylum level as well as many common taxa at the genus level. However, the HOMIM microarray method fails to detect approximately half of the bacterial species commonly present in saliva (*Ahn et al., 2011*).

In this study, we applied high-throughput sequencing (HTS) of the bacterial small-subunit ribosomal RNA (16S rRNA) genes to determine the salivary profiles of patients

with and without pancreatic cancer. The use of HTS to sequence 16S rRNA bacterial genes from entire salivary microbial communities allows for a more comprehensive profile of the microbiome in health and disease (*Kuczynski, Lauber & Walters, 2011*). During this study, we collected 146 saliva samples from patients at the UCSD Moores Cancer Center. HTS was used to characterize the salivary microbiome of patients with pancreatic cancer and compare them to patients with other diseases (including pancreatic disease, non-pancreatic digestive disease/cancer and non-digestive disease/cancer) as well as non-diseased (healthy) controls. This allowed us to test the hypothesis that patients with pancreatic cancer may have a distinct microbial community profiles compared to non-diseased controls and to other forms of digestive and non-digestive diseases. Our results demonstrated that patients with pancreatic cancer had a significantly higher abundance ratio of particular bacterial genera.

## MATERIALS AND METHODS

### Sample collection and patient information

This study was approved by the University of California San Diego (UCSD) and San Diego State University (SDSU) joint Institutional Review Board (IRB Approval #120101). Patients recruited for the study were being clinically evaluated at the UCSD Moores Cancer Center or were undergoing endoscopy procedures by UCSD Gastroenterologists in the Thornton Hospital Pre-Procedure Clinic between May 2012 and August 2013. All patients were required to fast for 12-hours prior to cancer evaluation and endoscopy procedures. To avoid bias during enrollment, the research coordinator responsible for recruiting participants was unaware of patient diagnosis at time of sample collection. Consenting participants were provided with IRB-approved consent forms, and HIPAA forms, as well as an optional, voluntary written survey in which they could share relevant information about antibiotic, dental and smoking history. All participants gave informed consent and their identities were withheld from the research team. Each subject was free to withdraw from the study at any time. Participants were asked to give a saliva sample into a 50 mL conical tube. If the amount of saliva exceeded 55 uL, 10 uL was transferred into tube containing Brain-Heart Infusion media (BHI) and glycerol for future culturing. The remaining saliva was broken up into 55 uL aliquots and stored in sterile cryovials. Both BHI and saliva samples were then immediately stored at −80 °C until further processing.

Of the 146 participants, three subjects voluntarily withdrew and seven were not included in the study due insufficient production of saliva (<55 uL) leaving 136 saliva samples. After sample collection, the research coordinator accessed the participants' medical records electronically for patient diagnosis information that was included under a novel subject ID number. Diagnosis was used to determine health status and assess the stage of disease when each sample was taken. The various diagnoses were grouped into the following categories: pancreatic cancer, other disease (including pancreatic disease, non-pancreatic digestive disease/cancer and non-digestive disease/cancer), and healthy (non-diseased) controls. Healthy individuals were defined as participants with no documented chronic digestive or non-digestive disease, and a 5-year resolution of any

previously documented digestive or non-digestive disease. Exclusion criteria included participants undergoing active chemotherapy or radiation therapy or use of antibiotics two weeks prior to saliva collection as well as invasive surgery in the past year.

## DNA isolation, PCR and 16S rRNA sequencing

Bacterial DNA was extracted directly from 50 uL of patient saliva using the MoBio PowerSoil DNA Extraction Kit (Catalogue 12888-05, Mo Bio Laboratories, Carlsbad, CA, USA) following the manufacturer's protocol. Genomic DNA was quantified using the NanoDrop$^{TM}$ Spectrophotometer and stored at $-20\,°C$.

The 16S ribosomal RNA (rRNA) amplicon region was amplified using barcoded 'universal' bacterial primer 515F (5′-AATGATACGGCGACCACCGAGATCTACAC TATGGTAATT GT GTGCCAGCMGCCGCGGTAA-3′) and 806R (5′-CAAGCAGAAGA CGGCATACGAGAT XXXXXXXXXXXX AGTCAGTCAG CC GGACTACHVGGGTWT CTAAT-3′) (X's indicate the location of the 12-bp barcode) with Illumina adaptors used by the Earth Microbiome Project (http://www.earthmicrobiome.org/emp-standard-protocols/16s). The barcoded primers allow pooling of multiple PCR amplicons in a single sequencing run. PCR was carried out using the reaction conditions outlined by the Earth Microbiome Project. Thermocycling parameters were as follows: 94 °C for 3 min (denaturing) followed by amplification for 35 cycles at 94 °C for 45 s, 50 °C for 60 s and 72 °C for 90 s, and a final extension of 72 °C for 10 min (*Caporaso et al., 2011*). PCR amplicons were then sequenced on the Illumina MiSeq platform at the Argonne National Laboratory Core sequencing facility (Lemont, IL).

## Sequence analysis

16S rRNA sequences were de-multiplexed using the Quantitative Insights Into Microbial Ecology (QIIME v.1.8.0, http://www.qiime.org) pipeline. Sequences were grouped into operational taxonomic units (OTUs) at 97% sequence similarity using the Greengenes reference database. OTUs that did not cluster with known taxa at 97% identity or higher in the database were clustered *de novo* (UCLUST (*Edgar, 2010*). Representative sequences for each OTU were then aligned using PyNast (*Caporaso, Bittinger & Bushman, 2010*), and taxonomy was assigned using the RDP classifier (Version 2.2) (*Cole et al., 2003*). A phylogenetic tree was built using FastTree (*Price, Dehal & Arkin, 2009*). Before performing downstream analysis, patient samples were rarefied to 100,000 sequences per sample, singletons and OTUs present in <25% of samples were removed prior to rarefaction. Chimeric sequences were identified using ChimeraSlayer in QIIME, as well as with DECIPHER (*Wright, Yilmaz & Noguera, 2012*), and subsequently removed. Alpha diversity metrics were computed using QIIME. Beta diversity distance between samples (weighted and unweighted UniFrac) were computed and used to account for both differences in relative abundance of taxa and phylogeny (*Vázquez-Baeza et al., 2013*). Beta diversity comparisons were done using analysis of similarities (ANOSIM). We also tested whether there were significant differences in abundance ratios of particular genera between our different categories with GraphPad Prism version 6.0 using the Kruskal–Wallis test followed by Dunn's multiple comparison correction. Statistical significance was accepted

at a $p < 0.05$. Analysis and identification of potential contaminants was done using SourceTracker (*Knights et al., 2011*).

## Quantitative PCR (qPCR)

*Leptotrichia* abundance was determined using qPCR. Briefly, for each sample we estimated *Leptotrichia* abundance using *Leptotrichia* specific 16S primers and normalized their values to overall bacterial abundance estimated using qPCR with universal bacterial 16S primers (5′-TCCTACGGGAGGCAGCAGT-3′ forward primer, and 5′-GGACTACCAGGGTATCTAATCCTGTT-3′ reverse primer) developed by *Nadkarni et al. (2002)*. qPCR was performed on a Bio-Rad CFX96 Touch™ Real-Time PCR Detection Instrument. The maximum $C_t$ (threshold cycle) for the universal 16S primers was set to 35 cycles and $C_t$ levels above this threshold were considered background noise. Genus-specific primers for amplification of *Leptotrichia* were designed using 16S rRNA sequences obtained from the RDP classifier (Version 2.2) (*Cole et al., 2003*). Primer3 online software was used for primer selection, and conditions were settled following the recommendations of *Thornton & Basu (2011)*. The *Leptotrichia* forward primer sequence (5′-GGAGCAAACAGGATTAGATACCC-3′) and the *Leptotrichia* reverse primer sequence (5′-TTCGGCACAGACACTCTTCAT-3′) generated an amplicon of 87 bp. The PCR reaction contained 1 uM of both forward and reverse *Leptotrichia* primers with thermocycling parameters of 50 °C for 2 min, 95 °C for 10 min and 40 cycles of 95 °C for 15 s and 62.5 °C for 1 min. The amplification reactions for the universal primers and *Leptotrichia* primers were carried out in at least duplicate using 25 uL of SYBR Green Master Mix (Bio-Rad) and 0.85 ng/uL of extracted DNA as template. Various online tools, including *In silico* PCR Amplification (*Bikandi et al., 2004*) and Ribosomal Database Project (*Cole et al., 2003*) were used to check the specificities of the oligonucleotide primer sequences for the target organism. A saliva sample was sequenced (Eton Bioscience, San Diego, CA) using our novel primers and primer specificity was further confirmed with a 16S rRNA database BlastN search.

## RESULTS

Salivary microbial diversity profiles were generated for a total of 108 patients. 8 patients were diagnosed with pancreatic cancer (P), 78 were diagnosed with other diseases (including cancer) (O), and 22 were considered healthy (non-diseased) controls (H). Table 1 details the individual clinical characteristics, including, gender and ethnicity. Of the 108 patients, 23 patients in pancreatic, digestive, and non-digestive disease categories were diagnosed with having cancer. Table 2 details the types of cancer, as well as category groupings and the mean age of the cancer patients in each category.

Illumina sequencing yielded approximately 6.8 million sequences across all samples. The sequences are available on FigShare (http://dx.doi.org/10.6084/m9.figshare.1422174) along with the mapping file (http://dx.doi.org/10.6084/m9.figshare.1422175). An analysis of potential sample contamination using SourceTracker (*Knights et al., 2011*) identified some evidence of human skin and/or environmental contamination. The sequences associated with OTUs identified as contaminants, mostly *Staphylococcus* (skin)

**Table 1  Clinical characteristics of study sample ($n = 108$).**

| Demographics | Pancreatic cancer (P) $n = 8$ | Other disease (O) $n = 78$ | Healthy control (H) $n = 22$ | Total $n = 108$ |
|---|---|---|---|---|
| Sex | | | | |
| Male | 6 | 38 | 12 | 56 |
| Female | 2 | 40 | 10 | 52 |
| Ethnicity | | | | |
| Caucasian | 6 | 56 | 15 | 77 |
| Hispanic | 2 | 6 | 5 | 13 |
| Asian | 0 | 4 | 1 | 5 |
| Unknown | 0 | 12 | 1 | 13 |

**Table 2  Types of identified cancers ($n = 23$).**

| Cancer by category | Age mean | N |
|---|---|---|
| Pancreas ($n = 8$) | 71.1 | |
| Pancreatic cancer | | 8 |
| Digestive ($n = 9$) | 64.7 | |
| Ampullary | | 3 |
| Esophageal | | 3 |
| Stomach | | 1 |
| Rectal | | 2 |
| Non-digestive ($n = 6$) | 54.8 | |
| Breast | | 1 |
| Skin | | 1 |
| Testicular | | 1 |
| Thyroid | | 3 |

and *Cyanobacteria* (chloroplasts), were removed from all subsequent analyses. From these data, we identified a total of 12 bacterial phyla and 139 genera. Proteobacteria, Actinobacteria, Bacteroidetes, Firmicutes, and Fusobacteria were the 5 major phyla, accounting for 99.3% of oral bacteria (Fig. 1). The mean relative abundance of Proteobacteria was lower in pancreatic cancer patients relative to other sample categories, while Firmicutes tended to be higher, though these were not significant after adjusting for multiple comparisons (FDR). The pancreatic cancer group also had higher levels of *Leptotrichia*, as well as lower levels of *Porphyromonas*, and *Neisseria* (Fig. 2). In general, multi-level taxonomic profiles of the healthy group resembled the 'other' disease group, while the pancreatic cancer group was readily distinguishable (Fig. S1). However, there were no significant differences among the three main groupings (H, O, and P) in either beta diversity (ANOSIM; $P = 0.1$) or alpha diversity (Chao1, K–W test; $P = 0.6$; Faith's PD, K–W test; $P = 0.56$).

As in previous studies by *Farrell et al. (2012)* and *Lin et al. (2013)*, we saw lower relative abundances of *Neisseria* and *Aggregatibacter*, although these differences were not

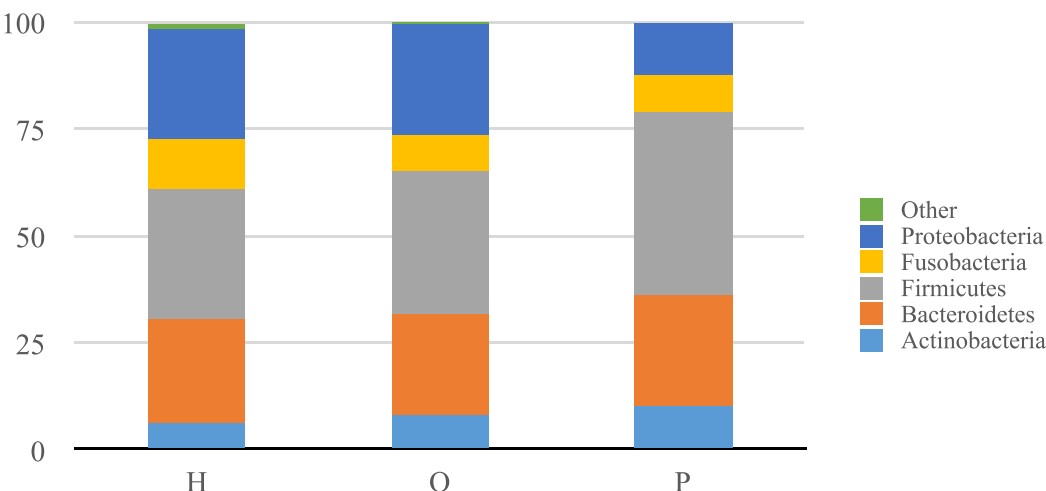

**Figure 1 Relative abundance of phyla identified in patient saliva summarized by diagnosis group.** Relative abundance of phyla in oral communities from 108 study patients summarized by diagnosis group (H, healthy control; O, other disease; and P, pancreatic cancer).

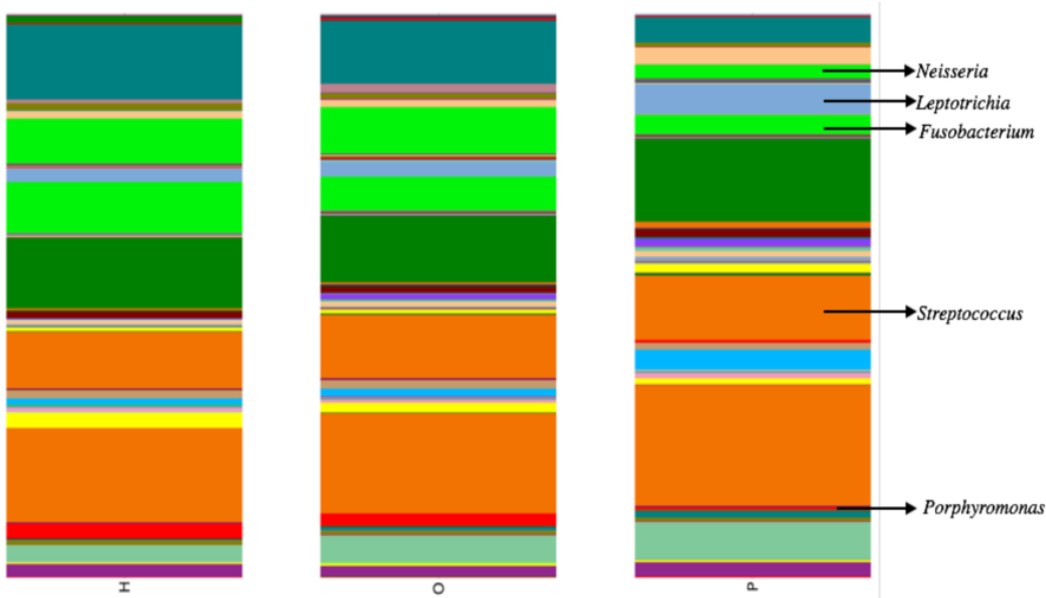

**Figure 2 Mean relative abundances of particular genera in pancreatic cancer patients (P) compared to healthy (H) and other disease (O) patient groups.** Relative abundances of genera in oral communities from 108 patients. Arrows point to specific genera that showed interesting trends across diagnosis groups.

significant (K–W test; $P = 0.07$ and $P = 0.09$ respectively). *Bacteriodes* was more abundant in pancreatic cancer patients compared to healthy individuals, similar to what Lin et al. observed, although this too was not significant (K–W test; $P = 0.27$). We did not see any difference in the relative abundance of *Streptococcus* or *Granulicatella*, which were shown to differ in a prior pancreatic cancer study (*Farrell et al., 2012*). Additional analytical targets were based on a preliminary study consisting of our first 61 saliva samples (including 3

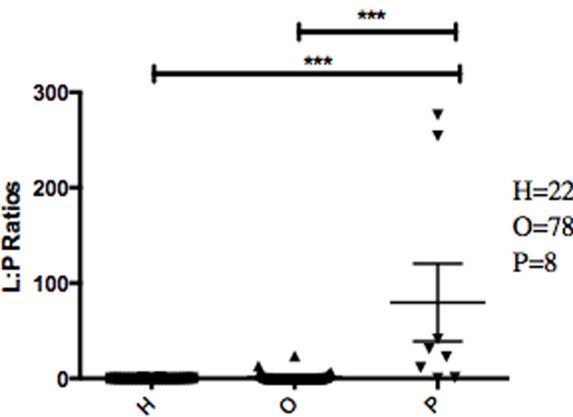

**Figure 3 Abundance ratio of *Leptotrichia* to *Porphyromonas* between different patient categories.** Each symbol represents the ratio of *Leptotrichia Oral Taxon 221* and *Leptotrichia hongkongenesis* to *Porphyromonas* for an individual patient ($n = 108$). Patients are grouped into 3 different categories depending on their diagnosis: healthy control (H), other diseases (including cancer) (O), and pancreatic cancer (P). Horizontal bar and error bars represent the mean and SEM, respectively. ***$p < 0.001$ (Kruskal–Wallis test followed by Dunn's multiple-comparison test).

from pancreatic cancer patients), which showed significantly higher *Leptotrichia* and lower *Porphyromonas* in pancreatic cancer patient saliva.

The abundance ratio of *Leptotrichia*, specifically two OTUs (arbitrarily named OTU 31235 and OTU 4443207), to *Porphyromonas* was significantly higher in pancreatic cancer patients (Fig. 3). A BLAST comparison of these OTUs to the 16S sequence in the Human Oral Microbiome database (*Chen et al., 2010*) (HOMD RefSeq Version 13.2) found OTU 31235 to be 100% similar to *Leptotrichia sp. Oral taxon 221,* while OTU 4443207 was 99.3% similar to *Leptotrichia hongkongensis*. We found a strong positive correlation (Pearson's correlation $r = 0.903$, $P = 0.0000001$) between *Leptotrichia* abundances obtained from 16S rRNA sequencing (OTU relative abundances) and from real-time qPCR (Fig. 4).

## DISCUSSION

Our analysis of salivary microbial profiles supports prior work suggesting that salivary microbial communities of patients diagnosed with pancreatic cancer are distinguishable from salivary microbial communities of healthy patients or patients with other diseases, including non-pancreatic cancers. At the phylum level, pancreatic cancer patients tended to have higher proportions of Firmicutes and lower proportions of Proteobacteria (Fig. 1). At finer taxonomic levels, we observed differences in the mean relative abundances of particular genera in pancreatic cancer patients compared to other patient groups (Fig. 2). For instance, there was a higher proportion of *Leptotrichia* in pancreatic cancer patients, while the proportion of *Porphyromonas* and *Neisseria* were lower in these patients.

The most striking difference we found between the microbial profiles of pancreatic cancer patients and other patient groups was in the ratio of the bacterial genera *Leptotrichia* and *Porphyromonas* (LP ratio) (Fig. 3). The LP ratio had been identified as a potential biomarker from a preliminary analysis and an analysis of the full dataset found significantly higher LP ratio in pancreatic cancer patient saliva than in other patient

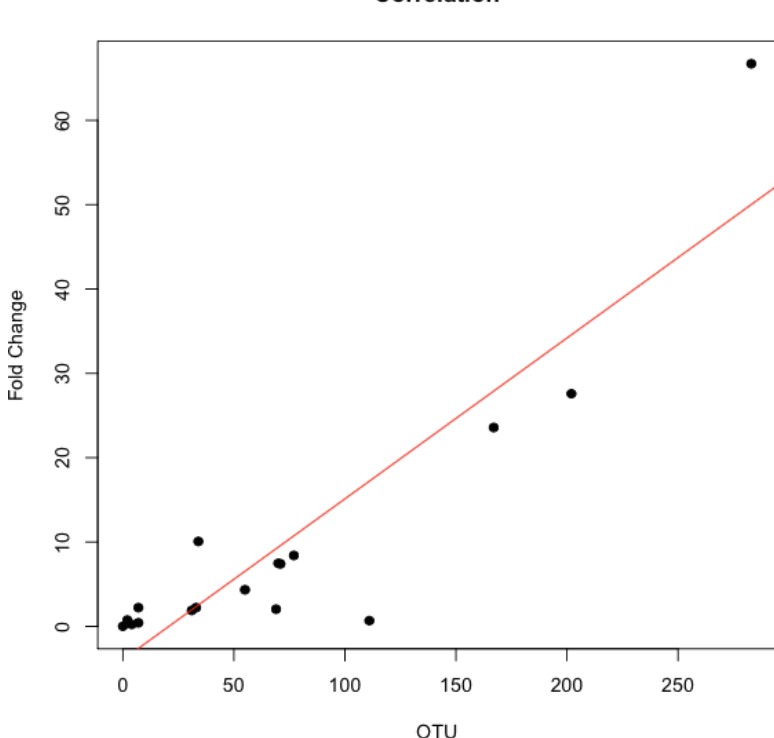

**Figure 4 Correlation between *Leptotrichia* abundance from 16S rRNA sequences and from real-time qPCR.** Cross validation of total *Leptotrichia* OTU abundance using real-time qPCR. After using 16S rRNA as a reference gene for normalization of the levels of *Leptotrichia* genus, data was normalized by fold change to three healthy controls with relatively low *Leptotrichia* OTU abundance. Each symbol represents a patient: P = 6, and O = 12. *Leptotrichia* OTU abundance was correlated with qPCR fold change according to Pearson's correlation ($r = 0.903$).

groups. To verify these differences using another method, we cross-validated the relative abundances of *Leptotrichia* (Fig. 4). Interestingly, during the analysis of the 16S rRNA data, we successfully used the LP ratio to reclassify one of the patients in the non-pancreatic cancer disease group. This particular individual had been initially diagnosed as having an unknown digestive disease, but the patient's high LP ratio suggested pancreatic cancer (Fig. 3). Subsequently, the patient was re-evaluated and diagnosed with pancreatic cancer, supporting the notion that the LP ratio may serve as a pancreatic cancer biomarker.

Despite the small cohort of patients in this study, we believe our results are especially noteworthy because we were able to distinguish between patients with pancreatic cancer and patients with a variety of other diseases (including non-pancreatic cancer), in addition to healthy controls. Other researchers have proposed the use of ratios of bacterial taxa previously. *Galimanas et al. (2014)* suggested using salivary bacteria abundance ratios as a means for differentiating between healthy and diseased patients. Taxonomic ratios have been used to differentiate between subjects in studies of obesity (*Lazarevic et al., 2012*), diabetes (*Zhang & Zhang, 2013*), and periodontal disease (*Moolya et al., 2014*). Ratio

comparisons also help to control for high levels of taxonomic variability among individuals (*Ding & Schloss, 2014*; *Segre, 2012*; *Schwarzberg et al., 2014*; *Wang et al., 2013*).

A review of the literature revealed that *Leptotrichia*'s role in oral health remains elusive. However, these bacteria have been found in the bloodstream of immune-compromised patients (*Eribe & Olsen, 2008*) and co-occur significantly with colorectal tumors (*Warren et al., 2013*). *Leptotrichia* have been isolated from cardiovascular and gastrointestinal abscesses, from systemic infections, and are thought to be pathogenic (*Han & Wang, 2013*). In regards to *Porphyromonas*, antibodies to *Porphyromonas gingivalis* have been directly associated with pancreatic cancer (*Michaud et al., 2012*). A European cohort study measured plasma antibodies to 25 oral bacteria in pre-diagnostic blood samples from 405 pancreatic cancer patients and 416 matched controls and found a >2-fold increase in risk of pancreatic cancer among those with higher antibody titers to a pathogenic strain of *P. gingivalis* (*Michaud et al., 2012*). At first glance, it appears contradictory that individuals with higher *Porphyromonas* antibody titers would have lower oral abundances. However, studies of systemic immunization of animals to particular periopathogens including *Porphyromonas* have shown reduced colonization of these bacteria in the mouth and a reduction of periodontitis (*Evans et al., 1992*; *Persson et al., 1994*; *Clark et al., 1991*). Similarly, higher *Porphyromonas* antibody titers in individuals with pancreatic cancer may decrease their oral abundance, though this connection needs to be formally tested.

Shifts in salivary microbial diversity could also be a systematic response to pancreatic cancer. Pancreatic cancer is known to weaken the immune system (*Von Bernstorff et al., 2001*), which could lead to overgrowth of oral bacteria and a shift towards systemically invasive periodontal pathogens. The proliferation of bacterial pathogens could assist cancer progression through systemic inflammation (*El-Shinnawi & Soory, 2013*) or immune distraction (*Feurino, Zhang & Bharadwaj, 2007*). Thus, an initial increase in *Porphyromonas* might be followed by a decrease due to systemic invasion and antibody production. Indeed, inflammation is thought to play a significant role in the development of pancreatic cancer (*Farrow & Evers, 2002*).

We also compared the relative abundances of several other bacterial genera that were indicated as potential biomarkers in previous work by *Farrell et al. (2012)*. Like Farrell et al., we found a lower proportion of *Neisseria* in pancreatic cancer patient saliva compared with the healthy and other disease category, though this trend was not significant. However, we did not find the same results as Farrell et al. for the other bacterial genera they identified. Our data also showed an increase in *Bacteroides* and decrease in the abundance of the bacterial genus *Aggregatibacter* in patients with pancreatic cancer, supporting the results of a pilot study by *Lin et al. (2013)*, though neither trends were significant.

Methodological differences between our study and the Farrell et al. study in particular, may partially explain our divergent results. For instance, the inability of the V4 region of the 16S rRNA gene to discriminate *Streptococcus mitis* from other *Streptococcus* species may have prevented us from detecting difference in this species' abundance (*Farrell et al., 2012*). Additionally, our study had a broader array of patient categories and cancers were not always confined to the pancreas at the time of sampling.

Interestingly, since the completion of our study, *Mitsuhashi et al. (2015)* reported the detection of oral *Fusobacterium* in pancreatic cancer tissue. A retrospective review of our abundance data also found a lower relative abundance of *Fusobacterium* in pancreatic cancer patients compared to other patient categories (Fig. 2; K–W test, $P = 0.03$ prior to FDR correction) suggesting the processes driving differences in *Fusobacterium* may be similar to our proposed mechanism for *Porphyromonas*. Although the result was not significant after adjusting for multiple-comparisons (FDR), we suggest *Fusobacterium* abundance should be considered as a potential biomarker target for future studies with larger patient cohorts.

Overall, our study suggests that members of the salivary microbiome have promise as potential pancreatic cancer biomarkers and we may have uncovered an important new prospect in this regard (i.e., the LP ratio). However, our relatively small number of samples from pancreatic cancer patients and the discrepancies between our findings and previous work indicate that much larger patient cohorts will be needed to determine whether salivary biomarkers are diagnostically useful. Future studies should focus on improved metadata collection, including diet and oral health information (i.e., periodontal disease), which would make it possible to run statistical analyses that control for multiple factors involved in shaping oral microbial diversity. It will also be important to sample the same individual's saliva over time to assess whether we can distinguish between disease stages and also to control for intra-individual variation. Further, it is possible that single biomarkers may never be able to consistently identify pancreatic patients from other conditions. Thus, we may need more complex metrics that combine the abundances of multiple salivary bacteria, metabolite profiles, and detailed patient metadata. Effective diagnostic biomarkers for pancreatic cancers have been difficult to find, but are sorely needed and have the potential to save thousands of lives each year.

## ACKNOWLEDGEMENTS

We thank S Owens and J Marcel for their help in our sequencing runs. We also thank M Wachter for assistance in enrolling patients into the study. Thanks to all the patients who volunteered to be a part of this study.

### Funding

This study was supported by NIH Public Health Service grants U54CA132384 and U54CA132379. EMF was supported by NIH R25GM058906. SMG was supported by an EPA STAR Graduate Fellowship and by NIH Training Grant 5T-32EB-009412. The funders had no role in study design, data collection and analysis, decision to publish, or preparation of the manuscript.

## Grant Disclosures

The following grant information was disclosed by the authors:

NIH Public Health Service: U54CA132384, U54CA132379.

NIH: R25GM058906.

EPA STAR Graduate Fellowship.

NIH Training: 5T-32EB-009412.

## Competing Interests

The authors declare there are no competing interests.

## Author Contributions

- Pedro J. Torres performed the experiments, analyzed the data, contributed reagents/materials/analysis tools, wrote the paper, prepared figures and/or tables, reviewed drafts of the paper.
- Erin M. Fletcher conceived and designed the experiments, performed the experiments, contributed reagents/materials/analysis tools.
- Sean M. Gibbons performed the experiments, analyzed the data, contributed reagents/materials/analysis tools, reviewed drafts of the paper.
- Michael Bouvet contributed reagents/materials/analysis tools, reviewed drafts of the paper.
- Kelly S. Doran conceived and designed the experiments, contributed reagents/materials/analysis tools, reviewed drafts of the paper.
- Scott T. Kelley conceived and designed the experiments, contributed reagents/materials/analysis tools, wrote the paper, prepared figures and/or tables, reviewed drafts of the paper.

## Human Ethics

The following information was supplied relating to ethical approvals (i.e., approving body and any reference numbers):

This study was approved by the University of California San Diego (UCSD) and San Diego State University (SDSU) joint Institutional Review Board (IRB Approval #120101).

## DNA Deposition

The following information was supplied regarding the deposition of DNA sequences:

The sequences and metadata were deposited in FigShare.

http://dx.doi.org/10.6084/m9.figshare.1422174

http://dx.doi.org/10.6084/m9.figshare.1422175.

## Supplemental Information

Supplemental information for this article can be found online at http://dx.doi.org/10.7717/peerj.1373#supplemental-information.

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
