# Peer review of "Characterization of the salivary microbiome in patients with pancreatic cancer"

_PeerJ, doi:10.7717/peerj.1373_

## Round 0.1 · original submission · Major Revisions

· Academic Editor

Major Revisions

Please carefully review and address all of the issues raised by the reviewers. Please provide a point by point response indicating how and where each issue is addressed in the revised manuscript.

Reviewer 1 ·

Basic reporting

The article is well-written and follows the structure expected for original research paper.

Experimental design

It appears that sequencing reads introduced by chimeras (chimeric reads) were not removed which could introduce substantial bias. Please address.

Multiple comparisons (multiple tests) have been conducted in the statistical analysis and no adjustments to the p-values have been conducted for these. This needs to be accounted for using statistical adjustment.

Validity of the findings

The number of cancer cases are very small in this study (n=8) which makes it very difficult to conduct statistical analyses comparing 4 other groups (pancreatic disease, health control, digestive disease, and non-digestive disease) across 139 genera; there is no way that any conclusions can be drawn as currently presented. The authors should compare pancreatic cancer cases to all other diseases and, perhaps, separately to healthy controls. There is simply not enough data to make all the comparisons presented. Furthermore, a strict approach should be taken to adjust for multiple comparisons; only robust statistical significant results will stand out and have the potential to be reproduced in other studies. Figure 2 is especially concerning given the choice of presentation of a ratio of two bacteria that were (likely by chance) statistically significant, or near significance, alone.

Additional comments

Line 283-please provide a reference for the “preliminary studies”; more justification is needed for calculating the abundance ratios between Leptotrichia and other bacteria-- this looks like a fishing expedition. How many ratios were tested?

Reviewer 2 ·

Basic reporting

In Figure 1, same/similar colors are used for multiple phyla, so it is difficult to interpret. One suggestion would be to only show distinct colors for the most abundant phyla, and then group all of the lower abundance phyla into an "Other" category.

Experimental design

There are no details provided about eating/drinking/mouthwash/toothbrushing prior to saliva sample collection. All of these factors can alter the salivary microbiome directly (i.e. bacteria in food or mouthwash killing bacteria) or indirectly (i.e. carbohydrate availability altering the bacterial community).
There is no significant metadata about the patients reported such as diet and lifestyle (i.e. smoker/non-smoker). These factors can also have substantial effects on the salivary microbiome.
It appears comparisons are only performed at the phylum or OTU level. The authors should perform comparisons at every level of the taxonomic tree.

Validity of the findings

The finding of an increased ratio of Leptotrichia to Porphyromonas is troubling. It seems that the raw number of OTUs rather than a relative abundance was used to determine this ratio; if this is the case, the ratio needs to be normalized for the total number of reads in each sample.
The human microbiome is impacted by many complex factors (diet, genetics/ethnicity, geography, age, lifestyle, etc.), and multivariate statistical methods should be adopted to help control for confounders. There are many multivariate tools available for this purpose. One such tool developed specifically for microbiome analysis applications is MaAsLin (http://huttenhower.sph.harvard.edu/maaslin).

Additional comments

The authors conduct a study of the salivary microbiome in a cohort of 6 pancreatic cancer patients, 22 healthy controls, 12 non-digestive disease patients, 13 non-cancer pancreatic disease patients, and 53 digestive disease patients. Because of the small sample sizes the study suffers from limited statistical power, however the authors report several significant associations that may be a valuable addition to the literature.

Additional comments:
Were differences in diversity (i.e. alpha-diversity) between groups considered?

---

## Round 0.2 · accepted · Accept

· Academic Editor

Accept

The issues raised by the reviewers have been adequately addressed.

Reviewer 2 ·

Basic reporting

No major changes since previous version.

Experimental design

No major changes since previous version.

Validity of the findings

No major changes since previous version.

Additional comments

My concerns have been addressed.